# Oral Palatability and Owners’ Perception of the Effect of Increasing Amounts of Spirulina (*Arthrospira platensis*) in the Diet of a Cohort of Healthy Dogs and Cats

**DOI:** 10.3390/ani13081275

**Published:** 2023-04-07

**Authors:** Davide Stefanutti, Gloria Tonin, Giada Morelli, Raffaella Margherita Zampieri, Nicoletta La Rocca, Rebecca Ricci

**Affiliations:** 1Department of Animal Medicine, Production and Health, University of Padova, Viale dell’Università 16, 35020 Legnaro, PD, Italy; 2Department of Biology, University of Padova, Via U. Bassi 58/b, 35151 Padova, PD, Italy; 3Vetekipp S.r.l., via del Cristo 326, 35127 Padova, PD, Italy

**Keywords:** Spirulina, *Arthrospira platensis*, dog, cat, nutraceutical, microalgae, palatability

## Abstract

**Simple Summary:**

Spirulina (*Arthrospira platensis*) is regarded as a functional food due to its valuable nutritional profile and components with potential antioxidant, anti-inflammatory, and immune-modulatory properties. However, the use of Spirulina in companion animals is still largely unexplored. Additionally, the palatability of nutraceuticals can be a primary factor in determining compliance with the protocol when they are used as complementary feed. The aim of this study was thus to assess the palatability and owners’ perception of the effects of increasing amounts of Spirulina tablets in client-owned dogs and cats for a 6-week-long trial. Overall, the animals well accepted the supplementation of Spirulina, with no significant side effect detected. Owners appeared willing to adopt dietary Spirulina supplementation in the future for their pets, especially owners of senior dogs.

**Abstract:**

The nutraceutical supplementation of Spirulina (*Arthrospira platensis*) in dogs and cats has not yet been investigated. The aim of this study was to evaluate if the dietary supplementation of increasing amounts of Spirulina for 6 weeks is palatable to pets and to assess the owner’s perception of such supplementation. The owners of the 60 dogs and 30 cats that participated in this study were instructed to daily provide Spirulina tablets starting with a daily amount of 0.4 g, 0.8 g, and 1.2 g for cats as well as small dogs, medium dogs, and large dogs, respectively, and allowing a dose escalation of 2× and 3× every 2 weeks. The daily amount (g/kg BW) of Spirulina ranged from 0.08 to 0.25 for cats, from 0.06 to 0.19 for small-sized dogs, from 0.05 to 0.15 for medium-sized dogs, and from 0.04 to 0.12 for large-sized dogs. Each owner completed a questionnaire at the time of recruitment and the end of each 2-week period. No significant effect on the fecal score, defecation frequency, vomiting, scratching, lacrimation, general health status, and behavioral attitudes was detected by the owners’ reported evaluations. Most animals accepted Spirulina tablets either administrated alone or mixed with food in the bowl. Daily supplementation of Spirulina for 6 weeks in the amounts provided in this study is therefore palatable and well tolerated by dogs and cats.

## 1. Introduction

*Arthrospira platensis*, also known as ‘’Spirulina’’, is a prokaryote microorganism of the cyanobacteria phylum [1]. Cyanobacteria are traditionally called blue-green algae due to their color, morphology, and photosynthetic abilities and are considered the evolutionary bridge between green plants and bacteria [2]. In recent years, the use of cyanobacteria as nutraceuticals has been given considerable attention in both the food industry and academic research due to their variety of nutritionally important compounds with a wide range of health benefits, such as proteins, polyunsaturated fatty acids, dietary fibers, and pigments such as chlorophyll *a*, *β*-carotene and *C*-phycocianin [2,3,4]. In particular, cyanobacteria *Arthrospira* and *Chlorella* dominate the microalgae nutrition market and, among the *Arthrospira* genus, the *A. platensis* and *A. maxima* species are the best known of those commercialized for food purposes [5,6].

The high nutritional profile of *Arthrospira* includes a protein content reaching 60–70% of dry weight with a profile that includes all the essential amino acids for dogs [7,8], whereas no study has evaluated the presence of taurine in Spirulina samples for making the same claim for cats. Spirulina lipidic content is 5–10%, with palmitic acid (C16:0), linoleic acid (C18:2, n-6), and *γ*-linoleic acid (C18:3, n-6) accounting together for over 80% of the total fatty acid content [9]. Spirulina is also a good source of various minerals, such as iron, magnesium, calcium, phosphorus, and vitamins, including vitamin B_12_ and vitamin E. The latter, together with *β*-carotene and phycobiliproteins such as phycocyanin and allophycocyanin, represent the primary antioxidant compounds contained in microalgae [2,10]. Studies conducted mostly on mice, rats, and humans have investigated the multiple beneficial effects of Spirulina, such as its antioxidant [10,11], anti-inflammatory [1,10], anti-diabetic [12,13], anti-viral [14,15], hypolipidemic [16,17,18], hepatoprotective [19], nephroprotective [20,21], cardioprotective [22,23], neuroprotective [24], and immunomodulatory properties [10,25]. Despite this, few studies have investigated the effects of Spirulina in dogs and cats. To the authors’ knowledge, an in vitro study conducted on macrophages isolated with bronchoalveolar lavage in cats demonstrated increased macrophage phagocytic activity when incubated with Escherichia Coli and exposed to a water-soluble extract of *A. platensis* [26], whereas a recent study performed in dogs showed an immune-stimulating activity of *A. platensis* when they were fed an extruded diet supplemented with 0.2% DM spray-dried Spirulina compared to the control group: dogs fed diets supplemented with *Spirulina* showed significantly higher vaccine response and higher levels of fecal IgA than the control group [27]. Another recent publication [28] assessed the effect of four edible microalgae, including *A. platensis*, in a canine gut in vitro mode; the authors concluded that the microbial saccharolytic activities and the shift in fecal bacterial composition were less pronounced than expected based on current literature in other species. The nutraceutical and therapeutical applications of Spirulina in companion animals are thus largely unexplored. For this reason, no recommendations are available in the literature on the amounts of Spirulina necessary to achieve health benefits in dogs and cats, nor is there substantial evidence of what such health benefits may be; at the same time, the risk of side effects is not yet known. Despite the lack of scientific studies on the potential effects of the inclusion of Spirulina in dog and cat food, its use has raised interest in the pet food market in recent years and led to rapid growth in the sale of pet food and supplements containing this cyanobacterium. Additionally, the palatability of such products has not been evaluated in pets. In the case of nutraceutical agents, palatability can have a major influence on the convenience of administration, which has an impact on both patient and owner compliance [29]. The authors’ hypothesis is that the palatability of Spirulina tablets in dogs and cats may be correlated with the daily amounts administered, with higher amounts leading to a decrease in palatability; additionally, the authors hypothesize that the owners’ evaluation of such supplementation will be positive and that the owners will not report negative changes in the health evaluation of their pet caused by Spirulina supplementation, such as increasing episodes of vomiting or diarrhea, and may instead notice positive effects, namely improvements in coat condition and physical activity levels of their pets. With this in mind, the primary aim of this study was to evaluate the oral palatability and owners’ perception of the effects of increasing amounts of *A. platensis,* administered in the form of tablets, in a population of client-owned dogs and cats. As the health benefits of Spirulina have been attributed to its high nutritional value and pigment concentration, a chemical analysis of the Spirulina used for the experimental period was considered of utmost importance before starting the trial and, therefore, a further objective of this present study. 

## 2. Materials and Methods

### 2.1. Animals 

Sixty privately owned dogs and thirty privately owned cats were enrolled in this study from October 2020 to May 2021. Animals were deemed suitable if they were adults (1–10 years old), had a BCS between 4 and 7, had not suffered from any recognized disease, had not been taking any medication for the previous 2 weeks, and had never taken Spirulina (*Arthrospira Platensis*). Only cats who lived exclusively indoors were recruited, and only owners with one single pet were allowed to participate in this study to avoid any interference during the supplement administration. Pregnancy and lactation were considered exclusion criteria. Each pet owner was required to sign an informed and written consent form prior to including their pet in the trial. This study protocol was approved by the Ethical Committee of the University of Padova (OPBA, University of Padova, and Italian Ministry of Health, no. 62/2020)

### 2.2. Intervention Protocol 

Animals received Spirulina (provided by Livegreen s.r.l.) supplementation for a period of 6 weeks in the form of 0.4 g tablets. Dogs were divided into 3 groups of 20 subjects, each based on size (small: 1–10 kg; medium: 11–25 kg; large: >25 kg), and owners were instructed to provide Spirulina daily for 42 days, starting with a daily amount of 0.4 g (1 tablet), 0.8 g (2 tablets), and 1.2 g (3 tablets) for small, medium, and large dogs, respectively. Then, they proceeded with a dose escalation of 2× and 3× every 14 days, equal to 0.06 to 0.19 g/kg BW for small dogs, 0.05 to 0.15 g/kg for medium dogs, and 0.04 to 0.12 g/kg BW for large dogs (Table 1). Cat owners followed the same instructions given to small dog owners, thus providing 0.4 g of Spirulina for the first 2 weeks, 0.8 g for weeks 3 and 4, and 1.2 g for the last 2 weeks, equal to 0.08 to 0.25 g/kg BW. The initial daily amount of Spirulina was selected after reviewing the literature concerning the use of Spirulina in humans, which is usually in the range of 1–4 g/day, equal to 0.01–0.06 g/kg/day [16,17,18].

### 2.3. Palatability Test

In order to test the palatability of Spirulina, a protocol aimed at assessing the voluntary acceptance of Spirulina tablets based on the “Guidelines on the demonstration of palatability of veterinary medicinal products” [30] was adopted; owners were thus given the following instructions: (a) to first offer the tablets alone in an empty bowl, without food, to verify the voluntary uptake; only in case the animal refused to consume Spirulina in this way after one minute, the tablets were to be administrated (in order) (b) with the meal, (c) hidden in a tasty bite, and (d) crumbled and mixed in the meal. When multiple tablets were to be administrated on the same day, owners were instructed to do so in a single administration. If the animal refused to consume the tablets under any method, they would be discontinued from the trial. If the animal had experienced side effects that could be traced to the supplementation (e.g., vomiting or diarrhea), owners were asked to continue the administration for three days from their onset; if these effects did not disappear in such period of time, supplementation would be discontinued. During the 6-week trial period, owners were not allowed to make any changes to their pet’s diet (i.e., add or remove an ingredient or use another nutraceutical supplement).

### 2.4. Impact of Spirulina Intake on Pet’s Health

Each owner was asked to fill out the same questionnaire created using the Google Forms^©^ application at the moment of recruitment (T0), day 14 (end of T1), day 28 (end of T2), and day 42 (end of T3). In total, the T0 survey contained 38 questions, the T1 and T2 surveys consisted of 24 questions, and the T3 survey had 27 questions. The T0 survey included multiple choice questions (MCQs) for demographic data on the owner (i.e., gender and age), pet signalment (i.e., species, breed, gender, age, size, body weight, and body condition according to the owner’s perception), and pet diet (i.e., type of diet, number of daily meals, and eating behavior). In all surveys, participants were questioned about specific health conditions of their pets. More specifically, regarding gastrointestinal health, the following information was retrieved: 1. defecation frequency, 2. fecal score (according to the Purina^®^ Fecal Scoring Chart), and 3. frequency of diarrhea and vomiting episodes. Information on coat condition, scratching, and lacrimation was also gathered to assess if Spirulina supplementation may affect these parameters. A dog’s coat’s condition was assessed as 1. poor and opaque, 2. normal, or 3. optimal and shiny. Scratching and lacrimation were rated on a 3-point scale, where 1 was normal, 2 was moderate, and 3 was abundant. Furthermore, Likert scale questions were used to acquire information on the pet’s physical activity level, behavioral attitudes, and general health status. In particular, levels of physical activity, sedentariness, liveliness, and calmness were assessed using a 5-point Likert scale, where 1 equaled “not at all” active/sedentary/lively/calm, and 5 equaled “very much” active/sedentary/lively/calm. At T0, owners were asked to answer questions regarding the 2 previous months; on days 14, 28, and 42, they were asked to reply to questions regarding the previous 2 weeks. Owners were also asked to report the number of missed supplementation days. The survey compiled at T0 featured an additional section with questions concerning the owner’s knowledge of the nutritional properties and application of Spirulina, while the T3 survey included a final section with MCQs on the general opinion of the owners on the Spirulina supplementation in their pet after the six week-trial. At the beginning of the questionnaires compiled at the end of T1, T2, and T3, an additional section examined whether the owner had faced any inconvenience in administering the Spirulina tablets to the dog or the cat; if so, the reason for the pet’s refusal was investigated.

### 2.5. Chemical Analysis

The Spirulina supplement used in this study (dried algae sampled from the raceway pond at the cultivation plant located in Arborea, Italy, managed by Livegreen Società Agricola srl) was analyzed by the CNX Laboratory of the Department of Animal Medicine, Production and Health of the University of Padua for proximate analysis and fatty acids (FA) profile and by the La-Chi Laboratory of the Department of Comparative Biomedicine and Food Science of the University of Padua for the amino acid profile and the quantification of minerals. Spirulina tablets were also sent to Merieux Nutrisciences Laboratory (Resana, Treviso, Italy) for the analysis of vitamins B12, C, and E and to the Photosynthesis and Plant Biotechnology Laboratory of the Department of Biology of the University of Padua for the quantification of pigments (Table 2).

#### 2.5.1. Proximate Analysis

The dry matter (DM) content was determined by oven-drying previously ground samples at 105 °C for 24 hours. Crude protein (CP) content was determined according to the Kjeldahl method (EC 152/2009—annex III method C) [31] and calculated using a nitrogen conversion factor of 6.25. Ether extract (EE) analysis was performed according to method H, procedure B’ reported in annex III of EC 152/2009 [31] using Soxhlet extraction with petroleum ether. The crude fiber (CF) was obtained by the Weende method according to the same regulation (EC 152/2009—annex III method I) [31]. Defatted samples were treated with boiling solutions of sulfuric acid and potassium hydroxide, and the residue was separated by filtration on sintered glass, washed, dried, weighed, and heated in a furnace at 475–500 °C. Weight loss after combustion was expressed as CF. The ash content was measured gravimetrically after combustion at 550 °C until white, light grey, or reddish ash was obtained and subsequently cooled to environmental temperature (EC 152/2009—annex III method A) [31]. 

#### 2.5.2. Fatty Acids Profile

The FA profile was determined as follows: samples were trans-methylated using a methanolic solution of H_2_SO_4_ (4%) in order to determine fatty acid methyl esters (FAME). A biphasic separation was obtained by adding 0.5 mL of distilled water and 1.5 mL of N-heptane to each sample. FAME were quantified by gas chromatography (Shimadzu GC17A) equipped with an Omegawax 250 column (30 m × 0.25 µm × 0.25 µm) and FID detector. Helium was used as the carrier gas at a constant flow of 0.8 mL/min. The injector and detector temperatures were 260 °C. Peaks were identified based on commercially available FAME mixtures (37-Component FAME Mix, Supelco Inc., Bellefonte, PA, USA). The results are expressed as % of total detected FAME.

#### 2.5.3. Amino Acids Profile

The amino acid profile was analyzed according to the methods described in the European Pharmacopoeia 5.0 (Concile of Europe, 2005) [32] by using an HPLC Agilent 1260 Infinity equipped with diode array and fluorescence detectors and an Agilent ZORBAX Eclipse AAA column (4.6 mm × 75 mm, 3.5 μm). Precolumn derivatization using o-phthalaldehyde (OPA) for primary amino acids and 9-fluorenylmethyl-chloroformate for secondary amino acids was performed. Precolumn derivatization of amino acids with OPA was followed by a reversed-phase HPLC separation. Because of the instability of the OPA–amino acid derivative, HPLC separation and analysis were performed immediately following derivatization. Fluorescence intensity of OPA-derivatized amino acids was monitored with an excitation wavelength of 348 nm and an emission wavelength of 450 nm. Precolumn derivatization of amino acids with 9-fluorenylmethyl chloroformate followed by reversed-phase HPLC separation with fluorometric detection was used. Each derivative eluted from the column was monitored by a fluorometric detector set at an excitation wavelength of 260 nm and an emission wavelength of 313 nm. Calibration of amino acid analysis instrumentation involved the analysis of the amino acid standard (Amino Acid Standard 0.1 nmol/μL 10/pk; Agilent Technologies, Santa Clara, CA, USA) consisting of a mixture of amino acids at a number of concentrations to determine the response factor and range of analysis for each amino acid. 

#### 2.5.4. Mineral Analysis

Mineral quantification (Ca; Fe; K; Mg; Na; P) was performed by the Inductively Coupled Plasma–Optical Emission Spectrometry (ICP-OES) method (Spectro EOP, Ciros Vision) after microwave digestion (Association of Official Analytical Chemists 2000, procedure 999.10) [33].

#### 2.5.5. Vitamin Quantification

The products were analyzed for the quantification of vitamin B12 according to the methodic MP 2347 rev. 2 2021 [34], quantification of vitamin C according to the methodic MP 1106 rev. 2 2020 [35], and quantification of vitamin E according to the Reg CE 152/09 All. IV Met. B [31].

#### 2.5.6. Pigments Quantification

Analysis was performed considering two separate fractions: lipid- and water-soluble pigments. For lipophilic pigment extraction, 50 mg of powdered *A. platensis* tablets were dissolved in 2 mL of N, N-Dimethylformamide and incubated in the dark at 4 °C for 24 h. Samples were centrifuged at 20,000 g for 5 min (Sigma laborzentrifugen 3k15, Sigma Laborzentrifugen GmbH, Osterode am Harz, Germany), and supernatant was analyzed using a spectrophotometer reading wavelengths from 350 to 750 nm. Pigments were quantified according to the equations from Moran [36] for chlorophyll a and from Chamovitz et al. [37] for carotenoids.
$$\mathrm{Chlorophyll}\;a\;(µg/\mathrm{mL})=A_{664}\times 11.92$$
$$\mathrm{Carotenoids}\;(µg/\mathrm{mL})=A_{461}-\left(0.064-A_{664}\right)\times 4$$

A_664_ and A_461_ are, respectively, absorbances at 664 and 461 nm. Concentrations were normalized for the weight of the samples. Phycobiliproteins were extracted starting from 10 mg samples using cold phosphate buffer (0.15 M NaCl, 0.01 M Na2HPO4) and glass beads (150–212 µm of diameter). Alternated cycles of bead beater and freezing were used to disrupt cells. Supernatant was collected and measured by spectrophotometer considering 350–750 nm wavelengths. Quantification was performed according to Bennet and Bogorad [38] with the following equations:$$\mathrm{Allophycocyanin}\;(\mathrm{mg}/\mathrm{mL})=\left[A_{652}-\left(0.208\times A_{615}\right)\right]/5.09)$$
$$\mathrm{Phycocyanin}\;(\mathrm{mg}/\mathrm{mL})=\left[A_{615}-\left(0.474\times A_{652}\right)\right]/5.34$$
where A_615_ and A_652_ are absorbance values at 615 and 652 nm, respectively. Concentrations were normalized by sample weight.

### 2.6. Statistical Analysis

The data collected from the survey were transferred to a spreadsheet (Excel, Microsoft) and subjected to descriptive analysis. Statistical analyses were performed with SAS 9.4 (SAS Institute Inc., Cary, NC, USA). Parametric data were compared with one-way analysis of variance (ANOVA), whereas Kruskal–Wallis was used for the non-parametric data; statistical significance was established if the probability of error was ≤0.05. Spearman’s rank correlation coefficient was used to assess relationships between the owners’ opinions on the Spirulina supplementation and the pet signalment data.

## 3. Results

### 3.1. Pet Owners’ Data

The majority of pet owners participating in this study were females (74.4%, n = 67/90) aged between 18 and 34 years old (46.7%, n = 42/90) (Table 3). Concerning the owners’ knowledge of Spirulina, the Internet was the main source of information (62.2%, n = 56/90), followed by friends and family (17.8%, n = 16/90); 5 owners (5.6%) did not know anything about Spirulina before participating in this study. A total of 4 owners were familiar with Spirulina because they had taken it personally (4.4%), while books were the main source of information for 3 (3.3%). The remaining owners (6.7%, n = 6/90) claimed they learned about Spirulina through other sources. At the beginning of this study, one-third of the owners (n = 30/90) said they had seen pet food containing Spirulina before, and 26.6% (n = 24/90) believed there were supplements containing Spirulina intended for pets in the market. 

### 3.2. Canine Population

Of the 60 participating dogs, 26 were mixed breed (43.3%); the remaining 34 (56.6%) belonged to 23 breeds and were divided as follows: 4 German Shepherds, 2 Golden Retrievers, Lessinia and Lagorai Shepherds, French Bulldogs, Shetland Shepherds, Shitzu, Border collies, King Charles Spaniel Cavalier, Cocker Spaniels, 1 Jack Russell Terrier, Italian Bracco, Italian Hound, Australian Shepherd, Flat Coated Retriever, Labrador Retriever, Boxer, Czechoslovak Wolfdog, English Bulldog, Pug, Maltese, American Shepherd Miniature, Weimaraner, and Dachshund. The average age of the dogs was 4.25 years. Most females (67.7%, n = 21/31) were sterilized, while most males (69.0%, n = 20/29) were intact (Table 4). The majority (71.6%, n = 43/60) of the dogs followed a predominantly dry commercial diet; 20.0% (n = 12/60) a homemade diet; 5.0% (n = 3/60) a predominantly wet commercial diet, and 3.3% (n = 2/60) a raw meat-based diet (e.g., BARF). Most owners (75.0%, n = 45/60) gave their dog 2 meals per day, 13.3% (n = 8/60) gave 3 daily meals, 8.3% (n = 5/60) provided only 1 meal, and 3.3% (n = 2/60) fed their dog ad libitum. Concerning the eating behavior of the animals, 43.3% (n = 26/60) of owners replied that when presented with a new food, their dog usually ate it immediately, 31.6% (n = 19/60) sometimes ate it and sometimes refused it, 23.3% (n = 14/60) always ate it immediately, and 1.7% (n = 1/60) usually refused it. When the dog was given tablets alone in an empty bowl, 28.3% (n = 17/60) of owners said that the dog sometimes ate and sometimes refused them, 26.6% of the canine population (n = 16/60) usually ate them immediately, 21.6% (n = 13/60) usually refused them, 16.6% (n = 10/60) always ate them immediately, and 6.6% (n = 4/60) always refused them.

### 3.3. Feline Population

The majority of the 30 cats recruited for this study were common European cats (80.0%, n = 24/30), while 20.0% (n = 6/30) belonged to another breed, namely 2 Maine Coons, 1 Scottish Fold, Persian, Manx, and Siamese. The average age of the cats was 6.5 years. All females were neutered (100%, 13/13), and only a minority of males (11.8%, 2/17) were intact (Table 5). At T0, the majority (56.6%, n = 17/30) of owners considered their cat’s body condition ideal, while 36.6% (n = 11/30) considered their cat slightly overweight. Only 1 person (3.3%) considered his cat slightly underweight, and another (3.3%) called his pet overweight. Most owners (70.0%, n = 21/30) said they fed their cat primarily a dry, kibble-based diet, while 30.0% (n = 9/30) offered their pet a wet diet. Almost half of the cats (46.6%, n = 14/30) were fed ad libitum, 33.3% (n = 10/30) were given 3 meals daily, 13.3% (n = 4/30) more than 3 meals, and 6.6% (n = 2/30) 2 meals per day. Half of the owners (n = 15/30) said that when a new food was presented to their cat, sometimes it was eaten, and sometimes it was refused; furthermore, 26.6% (n = 8/30) of the feline population usually ate it immediately; 20.0% (n = 6/30) always ate it immediately; 3.3% (n = 1/30) usually refused new foods. When a tablet was administered to the cat alone in an empty bowl, one-third (33.3%, n = 10/30) of owners said their cat sometimes ate it and sometimes refused it; 23.3% (n = 7/30) usually refused it; 20.0% of owners (n = 6/30) replied that their cat usually ate it immediately; 20.0% (n = 6/30) always refused it; finally, 1 participant (3.3%, n = 1/30) responded that the cat always ate it right away.

### 3.4. Dogs’ Spirulina Palatability Test and Owners’ Assessment

After the first 2 weeks of this study using the starting amount of Spirulina, 96.6% (n = 58/60) of owners said that administering Spirulina to their dog did not cause any problems, while 2 owners (3.3%) discontinued the supplementation because in 1 case the dog experienced diarrhea for 3 days from the start of the administration and its participation in this study was, therefore, discontinued. On the other, supplementation was discontinued on the tenth day for recurrent episodes of vomiting that continued even after supplementation was discontinued. During T2, in 93.1% (n = 54/58) of cases, Spirulina did not cause any clinical signs, while 2 owners (3.4%) discontinued the trial due to possible side effects. More precisely, 2 small dogs experienced vomiting for 3 consecutive days after the 2× increase in the initial daily amount. During the same experimental period, 2 owners (3.4%) discontinued the test due to the inability to properly follow the protocol because of personal logistic complications. Finally, during T3, the 3× increase in the initial amount of Spirulina caused the discontinuation of the trial for 2 dogs (3.7%): 1 medium-sized dog experienced diarrhea for 3 consecutive days, while another medium-sized dog discontinued the trial due to the onset of symptoms of pseudopregnancy. Overall, 86.6% (n = 52/60) of recruited dogs completed the 6-week trial without showing adverse effects. The modality of administration of the tablets adopted by dog owners throughout the trial is shown in Figure 1.

The graph shows that most dogs accepted the tablets offered alone in an empty bowl or mixed with the meal in the bowl, while only a minority of owners had to hide the tablets in a tasty bite or crumble them into the meal. The number of days of missed supplementation was recorded; at T1, 10.3% (n = 6/58) of dogs missed a day, 1.7% (n = 1/58) missed 3 days, and 1 dog (1.7%) missed 4 days. At T2, 5.5% (n = 3/54) of dogs missed 1 day, 1.8% (n = 1/54) 2 days, and 1.8% (n = 1/54) 4 days. At T3, 5.7% (n = 3/52) had not received spirulina for 1 day, while 1.9% (n = 1/52) missed 3 days of administration. No dog missed more than 4 days of supplementation overall. The dog’s general health status was described as “excellent” by 51.6% (n = 31/60) of owners at T0, 53.4% at T1 (n = 31/58), 59.2% at T2 (n = 32/54), and 67.3% at T3 (n = 35/52); as “good” by 43.3% (n = 26/60) of owners at T0, 44.8% (n = 26/58) at T1, 37.0% (n = 20/54) at T2, and 28.8% (n = 15/52) at T3; as “mediocre” by 5.0% of respondents (n = 3/60) at T0, 1.7% (n = 1/58) at T1, 1.8% (n = 1/54) at T2, and 3.8% (n = 2/52) at T3. No statistically significant effect on the dogs’ general health status was found among time periods (*p* = 0.531), and no change was detected in fecal score among periods (*p* = 0.669) (Figure 2).

The number of episodes of vomiting and diarrhea during the 3 time periods was: 0 for 43 and 48 dogs at T1, 51 and 43 dogs at T2, and 44 and 46 dogs at T3, respectively; equal to 1 for 12 and 7 dogs at T1, 3 and 8 dogs at T2, 4 and 4 dogs at T3; equal to 2 for 3 and 3 dogs at T1, 0 and 2 dogs at T2, 1 and 1 dogs at T3. No dog had more than three episodes of vomiting during the experimental period, and no dog had more than three episodes of diarrhea at T1, while this happened to a dog at T2 and T3. The number of episodes of vomiting (*p* = 0.904) and diarrhea (*p* = 0.837) did not differ over time. The dog’s coat was considered poor and opaque by 1 person at T0 (1.6%, n = 1/60), 2 at T1 (3.4%, n = 2/58), T2 (3.7%, n = 2/54), and T3 (3.8%, n = 2/52); it was considered normal by 71.6% of owners (n = 43/60) at T0, 58.6% (n = 34/58) at T1, 50.0% (n = 27/54) at T2, and 46.2% at T3 (n = 24/52); it was instead considered optimal and shiny by 26.6% of owners at T0 (n = 16/60), 37.9% at T1 (n = 22/58), 46.3% (n = 25/54) at T2, and 50.0% (n = 26/52) at T3. No significant time effect was identified in relation to dog hair condition (*p* = 0.112). At T0, scratching behavior was considered normal by 78.3% (n = 47/60) of owners, while 20.0% of owners considered it moderate (n = 12/60), and 1.6% (n = 1/60) abundant. At T1, 65.5% (n = 38/58) of owners considered their dog’s scratching normal; the same was true for 74.1% (n = 40/54) at T2 and 78.9% (n = 41/52) at T3. Scratching was moderate according to 34.5% of owners at T1 (n = 20/58), 25.9% (n = 14/54) at T2, and 21.1% (n = 11/52) at T3. According to the owners’ opinion, at T0, the lacrimation of their dog was defined as normal by 63.3% (n = 38/60), moderate by 35.0% (n = 21/60) of the respondents, and abundant by 1.7% (n = 1/60). During the test, most owners reported normal lacrimation, which was scored as such by 65.5% (n = 38/58) at T1, 63.0% (n = 34/54) at T2, and 71.2% (n = 37/52) at T3. Another substantial percentage of owners defined lacrimation as moderate, i.e., 34.5% (n = 20/58) at T1, 37.0% (n = 20/54) at T2, and 26.9% (n = 14/52) at T3. In the last 2 weeks of this study (T3), 1 owner reported abundant lacrimation (1.9%). There were no significant changes among time periods concerning scratching (*p* = 0.209) and lacrimation (*p* = 0.518). When owners were asked to rate their dog’s appetite at T0, considering the past 2 months, 56.6% (n = 34/60) considered it normal, 33.3% (n = 20/60) voracious, 8.3% (n = 5/60) capricious, and 1.6% (n = 1/60) poor. Their dog’s appetite was considered normal by 74.1% (n = 43/58) of the owners at T1, by 59.2% (n = 32/54) at T2, and by 61.5% (n = 32/52) at T3; a voracious appetite was identified by 20.7% (n = 12/58) of respondents at T1, by 33.3% (n = 18/54) at T2, and by 30.8% (n = 16/52) at T3; appetite was defined as capricious by 5.1% (n = 3/58) of the owners at T1, 7.4% (n = 4/54) at T2, and 5.8% (n = 3/52) at T3. Finally, 1 owner (1.9%, n = 1/52) reported a poor dog appetite in the last 2 weeks of this study (T3). There were no statistically significant changes over time in owner-reported scores for dogs’ physical activity (*p* = 0.742), sedentariness (*p* = 0.611), liveliness (*p* = 0.611), and calmness (*p* = 0.473). The average scores on the Likert scale are shown in Figure 3. 

### 3.5. Dog Owners’ Opinion on the Spirulina Supplementation

The overall impression of the owners who administered Spirulina to their dog for the 6 weeks of the trial was neutral for 75.0% (n = 39/52) and positive for 25.0% (n = 13/52). Most owners who completed the 6-week study (67.3%, n = 35/52) said they were likely to integrate Spirulina into their dog’s diet in the future. There was a positive correlation between the age of the animals and the owner’s intention to use Spirulina supplementation again in the future (ρ = +0.35): in fact, 84.0% of owners of dogs aged ≥ 6 years expressed this intention (n = 21/25), compared to 40.0% of owners of dogs aged 1 to 5 years (n = 14/35). Most owners (78.8%, n = 41/52), at the end of the 6 weeks, reported that they had not encountered any difficulties in administering Spirulina to their dogs. A total of 10 participants (16.6%) stated that their dog did not like the tablets, and 4 (7.7%) complained that the number of tablets in the last 2 weeks was too high.

### 3.6. Cats’ Spirulina Palatability Test and Owners’ Assessment

During T1, 1 owner (3.3%, n = 1/30) discontinued this study because he was not able to find a way to administer the tablet of Spirulina to his cat; in his opinion, the animal did not like the smell and taste of the tablet. The other participants (96.6%, n = 29/30) were able to effectively administer Spirulina to their cat in the first 2 weeks of this study. During T2, 17.2% (n = 5/29) of participants discontinued this study; 4 of these discontinued the trial because their cat refused to consume Spirulina in any way, and in 1 case, the owner decided to discontinue this study after the onset of auricular dermatitis which occurred 3 days after the increase from 1 to 2 daily tablets. The remaining owners (82.7%, n = 24/29) were able to administer Spirulina tablets to their animals during the entire third and fourth weeks. In the last 2 weeks, no participant discontinued this study: 80.0% of the cats initially recruited (n = 24/30) completed the 6 weeks of Spirulina supplementation, eating 3 tablets per day without problems. The modality of administration of the tablets adopted by cat owners throughout the trial is shown in Figure 4.

At T1, 10.3% (n = 3/29) of participants missed a single day of supplementation due to forgetfulness in 1 case and because the cat vomited shortly after intake in the other 2 cases. At T2, 12.5% (n = 3/24) missed 1 day of supplementation, while 4.2% (n = 1/24) missed 2. At T3, 12.5% (n = 3/24) missed 1 day, while 1 owner missed 3 days. No owner missed more than 3 days of supplementation overall. Cat health status was also scored as excellent, good, mediocre, or poor. At T0, 53.3% (n = 16/30) of owners rated their cat’s health as good and 46.6% (n = 14/30) as excellent. The general health status of the cats during the weeks of administration of Spirulina was excellent in 55.2% of cases at T1 (n = 16/29), in 50.0% at T2 (n = 12/24), and 66.6% of cases at T3 (n = 16/24); it was considered good by 44.8% of owners at T1 (n = 13/29), 45.8% (n = 11/24) at T2, and 29.1% (n = 7/24) at T3. In the last 4 weeks (T2 and T3), 1 owner (4.2%) defined the cat’s state of health as mediocre. There was no statistically significant change in overall health status among experimental periods and from the baseline (*p* = 0.693); the same was true for the fecal score (*p* = 0.613) (Figure 5).

The number of episodes of vomiting and diarrhea during the 3 periods of the Spirulina administration was 0 for 18 and 28 cats at T1, 15 and 22 cats at T2, and 12 and 24 cats at T3, respectively; it was 1 for 5 and 1 cats at T1, 4 and 0 cats at T2, 7 and 0 cats at T3; it was 2 for 6 and 0 cats at T1, 2 and 2 cats at T2, and 4 and 0 cats at T3. No cat experienced more than three episodes of diarrhea during the three experimental periods, and no cat had more than three episodes of vomiting at T1, while vomiting was reported in three cats at T2 and one cat at T3. The number of episodes of vomiting (*p* = 0.163) and diarrhea (*p* = 0.854) did not differ over time. For most owners (53.3%, n = 16/30) at T0, their cat’s hair was normal, 1 out of 3 (36.6%, n = 11/30) thought it was optimal and shiny, and 10.0% (n = 3/30) said it was poor and opaque after the first 2 weeks of supplementation (T1), 48.3% (n = 14/29) of owners defined their cat’s hair as optimal and shiny, 44.8% (n = 13/29) considered it normal, and 6.9% (n = 2/29) reported it as opaque. During the third and fourth weeks, 70.8% (n = 17/24) of owners still participating in the trial described an optimal and shiny coat, while the remaining 7 (29.2%) participants evaluated it as normal. At T3, 79.2% (n = 19/24) of owners reported optimal and shiny hair, while the remaining 5 (20.8%) described it as normal. Among the 24 cats that took Spirulina for the entire experimental period, a time effect on hair condition was detected (*p* = 0.003), with 29.2% (n = 7/24) defining the hair as optimal and shiny at T0, 50.0% (n = 12/24) at T1, 66.6% (n = 16/24) at T2, and 79.2% (n = 19/24) at T3. When asked to define the cat’s level of scratching at T0, 60.0% (n = 18/30) of owners assessed it as normal and 40.0% (n = 12/30) as moderate. Scratching during the test weeks was defined as normal by 55.2% (n = 16/29) of owners at T1, 62.5% (n = 15/24) at T2, and 54.2% (n = 13/24) at T3, moderate by 44.8% (n = 13/29) at T1, 37.5% (n = 9/24) at T2, and 45.8% (n = 11/24) at T3. Cat lacrimation at T0 was considered normal by 50.0% (n = 15/30) of owners, moderate by 46.6% (n = 14/30), and abundant by 1 owner (3.3%). Lacrimation was considered normal by 51.3% of participants (n = 15/29) at T1, 58.3% (n = 14/24) at T2, and 54.2% at T3 (n = 13/24); it was defined moderate by 48.3% (n = 14/29) of owners at T1, 41.6% (n = 10/24) at T2, and 45.8% (n = 11/24) at T3. No significant change for either scratching (*p* = 0.693) or lacrimation (*p* = 0.454) was detected among time periods. At T0, the appetite of most cats (70.0%, n = 21/30) was considered normal by their owners; in 5 cases (16.6%), it was judged voracious, and in 4 (13.3%), capricious. During this study, owners defined their cat’s appetite as normal (72.4%, n = 21/29, at T1; 54.2%, n = 13/24, at T2; 66.6%, n = 16/24, at T3), voracious (17.2%, n = 5/29, at T1; 20.8%, n = 5/24, at T2; 16.6%, n = 4/24, at T3). In the first 2 weeks of supplementation, 6.9% (n = 2/29) of owners said their cat’s appetite was poor; at T2, the same status was reported by 12.5% (n = 3/24) respondents and at T3 by a single owner (4.2%). Their cat’s appetite was capricious in the opinion of 1 owner at T1 (3.4%) and of 3 owners, both at T2 and T3 (12.5%). There were no statistically significant changes over time in owner-reported scores for physical activity (*p* = 0.617), sedentariness (*p* = 0.416), liveliness (*p* = 0.893), and calmness (*p* = 0.794) levels. The average scores on the Likert scale are shown in Figure 6. 

### 3.7. Cat Owners’ Opinion on the Spirulina Supplementation

At the end of the last questionnaire, owners were asked to express a conclusive opinion on the integration of Spirulina into their cat’s diet: this opinion was positive for 62.5% (n = 15/24) of owners whose cats had taken the tablets for all 6 weeks of the trial, and a neutral opinion was given by the remaining 9 owners (37.5%, n = 9/24). Most owners who completed the 6-week study (83.3%, n = 20/52) said they were likely to integrate Spirulina into their cat’s diet in the future. Among the cat owners who finished the trial, most said they had no difficulty administering Spirulina to their cat (66.6%, n = 16/24). Additionally, 1 in 4 owners (25.0%, n = 6/24) felt that the number of tablets to be administered at T3 was too high. A total of 4 owners (16.6%) said the main difficulty was that their pet did not like the tablets, while 1 participant (4.2%) felt that the tablets were too hard for the cat, who could not chew them easily.

## 4. Discussion

The purpose of this study was to evaluate whether Spirulina tablets prove to be palatable and well accepted by client-owned dogs and cats and if any side effects, such as gastrointestinal, dermatological, or behavioral effects, were identified by owners during a 6-week experimental period in which tablet dosage increased. Spirulina is listed among the substances that are “Generally Recognized as Safe” (GRAS) by the US Food and Drug Administration, and many toxicological studies have proven Spirulina’s safety for rats, mice, and humans [38,39,40,41,42]. Occasional mild side effects have been associated with Spirulina consumption in people, however, with insomnia and gastric problems arising most frequently [43]. Rare cases of severe side effects have also been reported in humans: 1 case of anaphylaxis as a consequence of the ingestion of a 300 mg Spirulina tablet [44], 1 case of rhabdomyolysis after the consumption of 3 g/day for 1 month [45], and 1 case of hepatotoxicity after the consumption of Spirulina for 5 weeks–in the latter case, neither the quantity nor the origin of the Spirulina supplement is known [46]. Furthermore, because cyanobacteria cells contain a high content of nucleic acids, high uric acid production can potentially occur in humans and other mammals from the metabolism of purines, giving rise to hyperuricemia and gout. For this reason, a maximum daily intake of 30 g is recommended in humans as a safety margin [47,48]. Moreover, while *A. platensis* does not produce any cyanotoxin, other species of cyanobacteria that proliferate in the same environmental conditions may synthesize hepatoxic and neurotoxic agents [49]; therefore, strict control of the production process is highly recommended to avoid the risk of biomass contamination [6]. Bautista et al. [50] reported an episode of hepatopathy in a dog following the consumption for three and a half weeks of a commercially available blue-green algae supplement (according to the label, 100% certified organic *Aphanizomenon flos aquae*): the supplement was submitted to toxicology testing which revealed the presence of hepatoxic microcystin produced by toxic cyanobacteria such as *Microcystis aeruginosa,* which proliferate in similar environmental conditions as *A. platensis*. Given the absence of studies concerning the effects of nutritional supplementation of Spirulina in pets, our study was intended to test the tolerability of Spirulina by showing the absence of side effects in dogs and cats following increasing dietary amounts of this microalgae. Additionally, a key factor in the owners’ ability to successfully administer supplements to their pets is their palatability [51]. To our knowledge, there are no studies in dogs and cats aimed at evaluating the palatability of Spirulina-based supplements, and palatability is considered a possible barrier to adoption in humans [52]. This was hypothesized to be related to Spirulina’s sensory attributes (taste and smell) that may be detrimentally affected when high amounts are used [53,54,55].

### 4.1. Dogs’ Spirulina Palatability Test and Owners’ Assessment

#### 4.1.1. Palatability Test

Among the few owners who chose to discontinue this study, not one discontinued the supplementation due to the complete refusal of the tablets by the animal. Palatability issues that are frustrating for pet owners are commonly known to arise, however. For this reason, we further investigated the method of administration that participants were obliged to adopt to make Spirulina tablets accepted by their pets. Among the methods of administration suggested to owners at the start of the experimental protocol, the presentation of the tablets alone in an empty bowl was the main form of administration used in all three phases, thus highlighting that the microalgae are generally palatable for the dog. Among the dogs that did not take the tablets alone in an empty bowl, it was sufficient to mix them with food in most cases; therefore, the administration was feasible for most owners. Nevertheless, one owner out of six reported that the dog did not like the tablets. Therefore, when considering the supplementation of Spirulina in the dog’s diet, the veterinarian should inform the owner that, in a minority of cases, dogs do not accept the microalgae supplied in tablet form. It cannot be excluded that Spirulina offered in forms other than tablets is more palatable to these dogs, however. According to the answer to the first questionnaire that investigated the pet’s eating behavior, most dogs that ate the Spirulina tablets only when hidden in a tasty bite or crumbled into the meal routinely refused any type of tablet. This suggests that the dogs’ refusal was not linked to Spirulina itself but rather to the form of the supplements instead.

#### 4.1.2. Gastrointestinal Signs

Spirulina did not cause any side effects in most dogs throughout the entire course of this study, even when double the initial daily amount was offered at T2 or even when three times the initial amount was given at T3. Only a small percentage of dogs (1/60 at T1, 2/58 at T2, and 1/52 at T3) had to discontinue the trial due to side effects probably related to the supplementation, 2 cases of vomiting and 2 of diarrhea. Considering gastrointestinal signs, in light of the data presented here, it can be stated that Spirulina supplied in tablet form is well tolerated by the canine species and that only in rare cases does it lead to the onset of episodes of vomiting or diarrhea. It should be noted that the frequency of such episodes was low in this study, even at the highest daily amounts. Moreover, supplementation was discontinued after three days of symptoms, and therefore, it cannot be excluded that gastrointestinal signs may have disappeared if the animals had continued consuming the tablets. In addition, no significant differences were identified among the experimental periods or compared to the baseline, neither regarding fecal score nor the number of episodes of vomiting and diarrhea. Therefore, the six-week administration of Spirulina in the amounts used in this study did not cause significant alterations related to these parameters in dogs. It is possible, however, that microalgae provide support to gastrointestinal functions in pets, given that they have been shown to beneficially modulate the composition of the intestinal microbiota in different species [25,56,57].

#### 4.1.3. Effects on Coat

The percentage of owners who described their dog’s hair as “optimal and shiny” increased substantially at the end of the experimental period (T3) compared to the baseline. Although this trend was not statistically significant, it is possible that a larger sample size or a longer intake period will confirm the positive effects on the skin and coat of pets. Spirulina has been amply reported in the literature to offer a valid support to the health of human skin and can thus be applied in the field of cosmetics [58]. In particular, *A. Platensis* has been shown to provide considerable benefits for the fibers composing the hair, making them brighter and easier to comb [59]. The benefits that Spirulina brings to the health of skin appendages probably derive from its rich nutritional profile because it has been shown that foods rich in antioxidants and omega-6 fatty acids can promote the protection of hair from oxidative stress derived from exposure to UV rays and environmental pollutants, improving shine and counteracting fall [60,61]. Furthermore, the excellent amino acid profile of the protein contained in *A. Platensis* may favor the composition of keratin of which human and animal hair are made. Other parameters evaluated in this study were scratching and lacrimation. Any variations in their manifestation are among the easiest signs for owners to notice; no significant variation between times was found in relation to these parameters.

#### 4.1.4. Effects on Dog Behavior

There were no significant changes in physical activity, liveliness, sedentariness, and calmness scores following Spirulina supplementation. These questions were asked because, in other species, it has been observed that the intake of Spirulina can positively affect parameters regarding physical activity. In humans, specifically, microalgae have proved effective in improving athlete sports performance thanks to their antioxidant activity [62]. A recent study carried out in rats also reported the effect of Spirulina in counteracting the fatigue of physical exercise [63]. The fact that, in this study, the daily assumption of increasing amounts of Spirulina did not affect such behavioral aspects is seen as positive, even if it does not allow the authors to draw definite conclusions. Only future studies that apply direct measures such as behavioral observations by trained personnel instead of indirect owner-reported assessment and subject animals to physical trials will enable the retrieval of more accurate information.

### 4.2. Cat Spirulina Palatability Test and Owner Assessment

#### 4.2.1. Palatability Test

Spirulina was generally palatable to cats: in all three phases of this study, the two most frequently used methods of administration were offering the tablets alone in an empty bowl or mixing intact tablets with food in a bowl. Although initially, most cats took Spirulina tablets alone in an empty bowl, probably also driven by a curiosity about the new food, this method was less welcome during phases T2 and T3; in addition to the fading of the novelty effect, the increase in the number of tablets to 2 and 3 per day could explain the decline in palatability and the consequent need to find a more acceptable method of administration. Hence, 16.6% of cat owners discontinued this study because they were unable to make their cats eat the tablets in any way. 

#### 4.2.2. Gastrointestinal Signs

No gastrointestinal side effects were noted by cat owners during the entire experimental period. No significant changes in the fecal score or the number of diarrheal or vomiting episodes were registered. Spirulina can thus be considered a well-tolerated feed supplement for this species when provided for 6 weeks at amounts ranging from 0.4 g to 1.2 g daily. 

#### 4.2.3. Effects on Coat

Interestingly, a significant increase in coat shine was detected in cats over time. Among owners who administered Spirulina continuously for the 6-week trial, those who scored the coat of their pet as optimal and shiny increased more than twice between the beginning and the end of the trial. This finding is in accordance with observations of the reactions of humans to the properties of Spirulina, which made their hair shinier and easily detangled [59], increased skin health, and visibly improved hair quality [61]. The number of owners that judged the condition of the coat as optimal and shiny also increased from 50% at the end of T1 to almost 80% at T3: a result that may be correlated to the increased amounts of Spirulina provided during T2 and T3. As it was for dogs, no significant effect on scratching or lacrimation was found in cats.

#### 4.2.4. Effects on Cat Behavior

As observed for dogs, there were no significant effects of the Spirulina supplementation on the cat behavior-related parameters assessed by the owners. However, in the open-ended question of the final survey, some owners reported that their cats seemed more active and playful during this study period. A recent study that evaluated the protective effect of dietary Spirulina supplementation against the impacts of monosodium glutamate in blood and behavior of Swiss mice showed that those belonging to the group supplemented with spirulina were more active compared to the control group [64]. Another study conducted on rats suggested that Spirulina may reduce anxiety and anhedonic behavior and ameliorate cognitive dysfunction [65]. In a randomized controlled trial conducted in humans, Spirulina supplementation was associated with a significant reduction in stress score [66]. Overall, for Spirulina and other nutraceuticals, evidence supporting the role of nutritional supplements and dietary ingredients in cognitive and behavioral health has grown over the last few years [67]. Thus, behavioral nutraceuticals may become a valid addition to behavior modification and environmental management plans [68]. This could be of particular interest for cats since, in recent decades, they have been removed from a free-roaming, active existence to a captive, indoor, sedentary condition, a transition that has led to a longer life expectancy but also to an increase in chronic health problems, including obesity and behavioral problems [69]. Further studies are necessary to investigate whether Spirulina can be useful in addressing these new challenges for cats’ well-being.

### 4.3. Owners’ Opinions on the Spirulina Supplementation

Regarding dogs, the owners of older subjects (aged six years or older) were seen to be more inclined to integrate the microalgae again into the diet of their pet, whereas this intention was decidedly less present among people who owned a young dog. This difference might possibly be due to the strong motivation owners have in supporting the vital functions of their senior dogs through the use of food supplements. In this sense, supplementing nutraceuticals with antioxidant and anti-inflammatory properties, such as Spirulina, seems to arouse interest and consensus, especially in this category of owners. Spirulina is known to counteract oxidative damage, inflammation, and immunosenescence, typically present in old age [70]. In humans, it has been shown that microalgae can play a key role in preventing age-related diseases, in particular, immunosenescence [71]. The same considerations may be applied to the canine species, given that the underlying pathophysiological mechanisms are often the same.

### 4.4. Limitations of this Study

The amounts of Spirulina used in this present study ranged from 0.04 (T1) to 0.19 g/kg/day (T3) for dogs and from 0.08 (T1) to 0.25 g/kg/day (T3) for cats. In the last phase of this study, the Spirulina intake adopted both for dogs and cats was considerably higher than usually suggested for humans, which is in the range of 1–4 g/day, equal to 0.01–0.06 g/kg/day [16,17,18,72,73]. However, our study demonstrated that no significant side effects were caused either in dogs or in cats, even at higher intakes. Further studies should be carried out to enable recommendations for the long-term use of Spirulina in dogs and cats in terms of both potential benefits and effective daily amounts. Given the lack of data on the use of Spirulina in dogs and cats and the fact that most of the health claims regarding its supplementation are unsubstantiated for these species [74], such trials are urgently required. This present study has two main limitations: the absence of a placebo-controlled group with owners blinded to the treatment and the owner-reported nature of the results, which may have been biased both by the lack of the placebo group and by the subjective nature of the reports. Since the objective of this study was to evaluate potential side effects arising from the intake of increasing amounts of Spirulina and to investigate the palatability of Spirulina tablets, priority was given to recruiting a significant number of dogs and cats for testing the supplement.

## 5. Conclusions

This study shows that since no noteworthy adverse effects were observed, a daily administration of *A. platensis* for 6 weeks in amounts ranging from 0.4 to 3.6 g/day in dogs and from 0.4 to 1.2 g/kg/day in cats is well tolerated. Additionally, Spirulina tablets appeared generally well accepted by both dogs and cats, and their palatability allowed administration alone in an empty bowl or mixed with food in the bowl. Nonetheless, a minority of animals, especially cats, may refuse them in any case despite the method of administration adopted. Owners appeared willing to adopt dietary Spirulina in the future for their dogs, especially owners of senior subjects. Further studies are needed to investigate the possible benefits of nutritional supplementation of Spirulina in pets.

## Figures and Tables

**Figure 1 animals-13-01275-f001:**
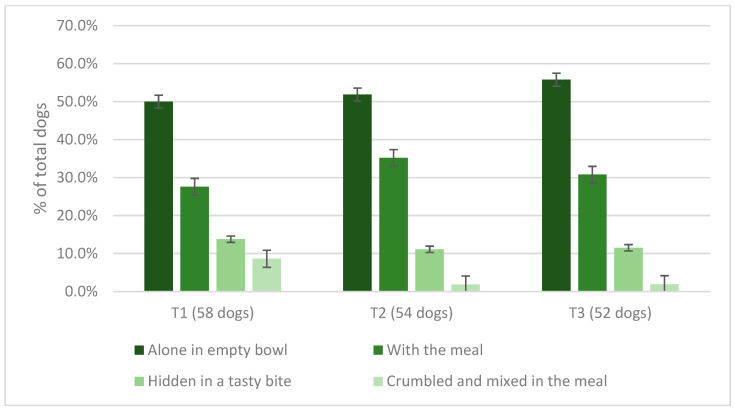
Method of offering of the increasing amounts of Spirulina offered to dogs at T1, T2, and T3 of this study.

**Figure 2 animals-13-01275-f002:**
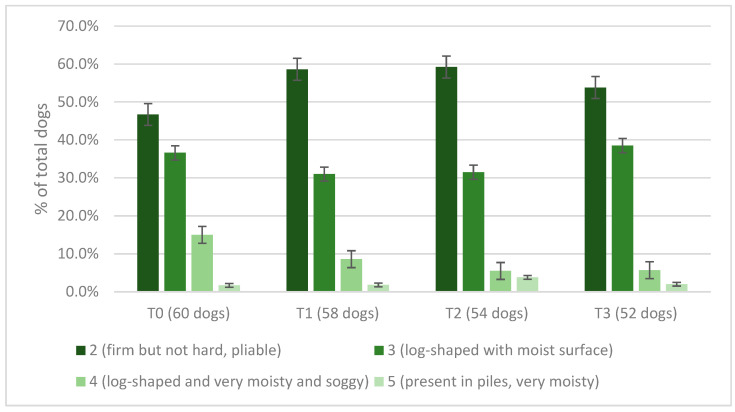
Fecal score of dogs according to owners at T0, T1, T2, and T3 (no owner reported a fecal score equal to 1, 6, or 7).

**Figure 3 animals-13-01275-f003:**
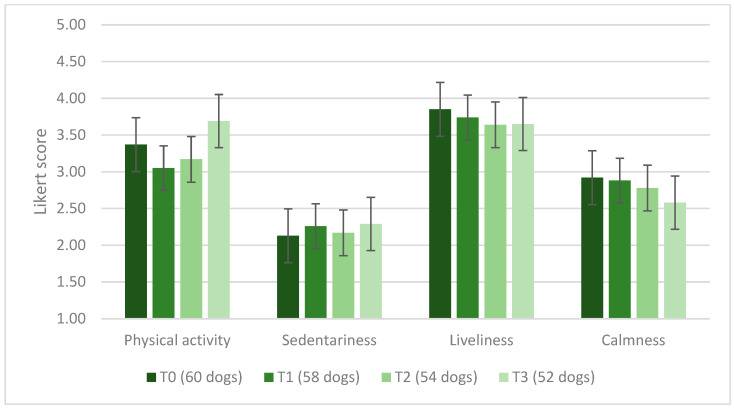
Average score on a Likert scale from 1, equal to “not at all”, to 5, equal to “very”, assigned by the owners evaluating the degree of physical activity, sedentariness, liveliness, and calmness of their dogs.

**Figure 4 animals-13-01275-f004:**
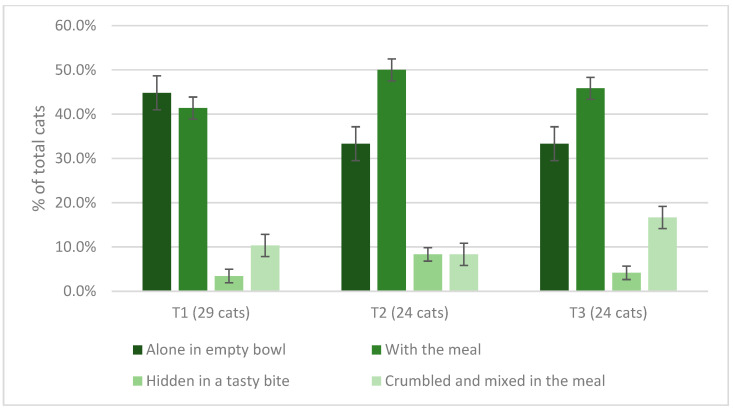
Method of offering of the increasing amounts of Spirulina offered to cats at T1, T2, and T3 of this study.

**Figure 5 animals-13-01275-f005:**
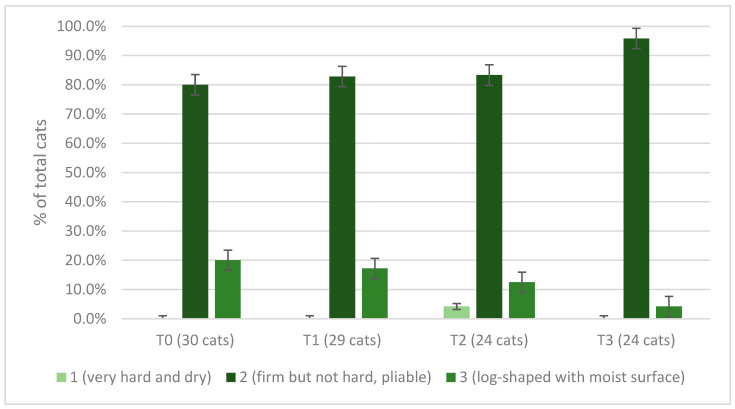
Fecal score of cats according to owners at T0, T1, T2, and T3 (no owner reported a fecal score higher than 3).

**Figure 6 animals-13-01275-f006:**
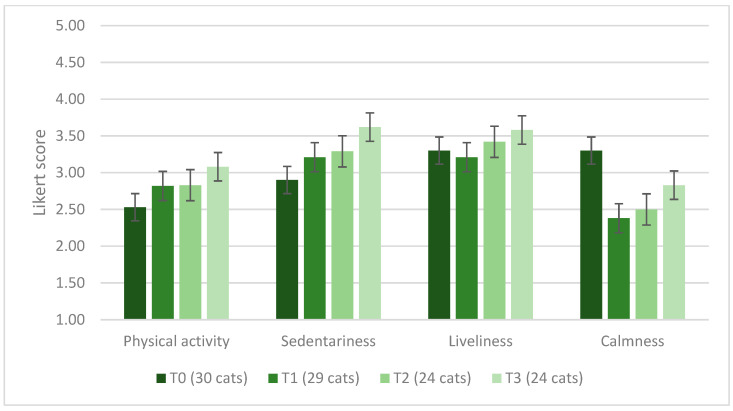
Average score on a Likert scale from 1, equal to “not at all”, to 5, equal to “very”, assigned by the owners evaluating the degree of physical activity, sedentariness, liveliness and calmness of their cats.

**Table 1 animals-13-01275-t001:** Daily amount (mg/kg BW) of Spirulina administrated to small dogs (*n* = 20), medium dogs (*n* = 20), large dogs (*n* = 20), and cats (*n* = 30) during the 42 days of the trial; data are expressed as means ± DS.

Animals	T1 (Days 1–14)	T2 (Days 15–28)	T3 (Days 29–42)
Small Dogs	64.9 ± 27.9	129.7 ± 55.8	194.6 ± 83.7
Medium Dogs	49.3 ± 15.0	98.6 ± 29.9	147.9 ± 44.9
Large Dogs	38.4 ± 5.0	76.8 ± 10.0	115.2 ± 15.1
Cats	83.7 ± 23.5	167.3 ± 47.1	251.0 ± 70.6

**Table 2 animals-13-01275-t002:** Nutritional composition of the Spirulina tablets used in the trial, including proximate analysis, fatty acid profile, amino acid profile, mineral and vitamin composition, and pigment content.

Nutrients	% DM ^1^
Crude Protein	60.44
Crude Fat	6.75
Crude Fiber	0.00
Ash	11.48
C14:0	0.03
C14:1	0.07
C15:0	0.03
C16:0	2.68
C16:1	0.27
C17:0	0.01
C17:1	0.02
C18:0	0.08
C18:1 Ω9	0.26
C18:2 Ω6	1.59
C18:3 Ω6	0.83
C18:3 Ω3	0.01
C20:0	0.01
C20:1 Ω9	0.01
C20:2	0.01
C20:3 Ω6	0.02
C20:5 Ω3	0.00
C22:6 Ω3	0.00
Histidine	1.27
Arginine	4.26
Serina	3.23
Glycine	2.87
Asparagine	5.55
Glutamine	9.93
Threonine	3.23
Alanine	4.58
Proline	2.37
Lysine	3.19
Methionine	0.50
Tyrosine	2.33
Valine	3.63
Cysteine	0.61
Isoleucine	3.18
Leucine	5.87
Phenylalanine	2.99
Taurine	0.07
Tryptophan	1.44
Calcium, Ca	0.46
Iron, Fe	0.05
Potassium, K	2.47
Magnesium, Mg	0.43
Sodium, Na	1.46
Phosphorus, P	1.35
Vitamin B12 (cyanocobalamin)	13.1 ± 2.6 ^2^
Vitamin C	<20 ^3^
Vitamin E	31.1 ± 7.8 ^3^
Chlorophyll *a*	1.6 ± 0.3 ^4^
Carotenoids	0.3 ± 0.1 ^4^
Allophycocyanin	21.9 ± 2.3 ^4^
Phycocyanin	50.9 ± 3.4 ^4^

^1^ Moisture level was equal to 6.86%. ^2^ Data expressed as μg/100 g DM. ^3^ Data expressed as mg/kg DM. ^4^ Data expressed as mg/g DM.

**Table 3 animals-13-01275-t003:** Demographics of owners (*n* = 90).

		Dog Owners, *n* (%)	Cat Owners, *n* (%)
Gender	Male	19 (31.7%)	4 (13.3%)
	Female	41 (68.3%)	26 (86.7%)
Age	18–34	25 (41.6%)	17 (56.7%)
	35–49	17 (28.3%)	10 (33.3%)
	50–64	15 (25.0%)	2 (6.7%)
	≥65	3 (5.0%)	1 (3.3%)

**Table 4 animals-13-01275-t004:** Characteristics of dogs enrolled in this study (*n* = 60).

		Dogs, *n* (%)
Gender	Male	29 (48.3%)
	Female	31 (51.7%)
Neutering Status	Yes No	30 (50.0%) 30 (50.0%)
Body Condition (According to Owner)	Underweight Ideal weight	3 (5.0%) 46 (76.7%)
	Overweight	11 (18.3%)

**Table 5 animals-13-01275-t005:** Characteristics of cats enrolled in this study (*n* = 30).

		Cats, *n* (%)
Gender	Male	17 (56.7%)
	Female	13 (43.3%)
Neutering Status	Yes No	28 (93.3%) 2 (6.7%)
Body Condition (According to Owner)	Underweight Ideal weight	1 (3.3%) 17 (56.7%)
	Overweight	12 (40.0%)

## Data Availability

The data presented in this study are available on request from the corresponding author.

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
