# Peer review of "Oral Palatability and Owners’ Perception of the Effect of Increasing Amounts of Spirulina (Arthrospira platensis) in the Diet of a Cohort of Healthy Dogs and Cats"

_animals, 2023, doi:10.3390/ani13081275_

Round 1

Reviewer 1 Report

I have read the manuscript and have some comments before the final decision of acceptance or rejection of the manuscript. So, some concerns need to be addressed as follows:

Major concerns

1) How did you calculate the sample size for the data included in this study? In my opinion, data from questionnaire-based studies require more samples than that, especially since the accuracy of evaluation varies from person to person.

2) The daily amount of spirulina is not fixed per kg of the body weight for all animals in the same category, but it is a range, and this isn't easy to give a valid recommendation.

3) The evaluation of palatability alone is insufficient to determine dose appropriateness. Instead, blood samples had to be taken, for example, at the beginning and end of the experiment, to assess the animals' health and correlate the findings with the chemical analysis of the spirulina.

4) The authors had to justify on what basis they selected the level of the tested spirulina. 

5) The significant differences between groups are not shown in the tables, figures, and text such as the p-value.

Additional comments:

1- Lines 22-23: The aim does not represent the experimental design of the study accurately. Please rewrite.

2- The study's objective should be clearly presented at the end of the introduction section. Also, the hypothesis of the study should be clarified before the objective.

3- Line 106: Spirulina source (product, concentration, company, city, country) should be clearly stated.

4- The breeds of dogs and cats should be mentioned.

5- Lines 174, 176, 178, 183, 221, and throughout the materials and methods section: add appropriate reference to the methods used

6- add a reference to the used palatability test.

7- Line 217: If a reference is made to AOAC, the specific procedure identification number must be mentioned.

8- Line 227: You should give this centrifugation value with x g and add the model of the used centrifuge and the producer company name and its location.

9- Line 228: “wavelengths from 350 to 750 nm” these values were different from what was reported next. 

10- Table 1: It is preferable to display the doses in "mg" instead of "g" as these are minor quantities.

11- In all figures, add the standard error bar.

12- Some references are cited in the text and are not included in the references list such as in L197, 217, 229, 230.

Author Response

Thank you for your review; it was very useful to improve this manuscript. We revised accordingly where possible, and here we try to further address your concerns.

This was a qualitative study realized to gather preliminary data on the nutraceutical use of Spirulina in pet nutrition. As it is specified in the study limitations, this study did not used a placebo group and the data were all owners-reported. We agree with you that it is not possible to make objective inference concerning Spirulina properties based on these data, but this was not our purpose; it is not possible (nor it was our intention) to evaluate the nutraceutical effects of Spirulina in pets with the present study. Our aim here was to develop some initial understanding about the potential use of this microalgae in pet nutrition, since there was no information in the literature concerning this subject. Would dogs and cats eat these tablets at all? In which amount? What would be the opinions of the owners? Would they notice possible side effects? These were the questions we were trying to answer, so that in the future further studies, especially randomized controlled trials, could be realized on this subject.

1-Since the qualitative and observational nature of this study, we did not perform sample size calculation as you would for a randomized controlled trial. Our aim was to recruit as many participants as possible overcoming the sample size of similar studies in the literature and what is suggested in the “Guideline on the demonstration of palatability of veterinary medicinal products” (Ema, 2014), which suggests a sample size of at least 50 animals if the product is administered only once, or at least 25 animals if the product is administered more than once, as it was in our case. Our opinion is that reaching a size of 60 dogs and 30 cats was thus an acceptable result, even when compared to similar trials recently published. For example, in 2022 Animals published a study (Berk et al.) on the oral palatability of MCT in dogs, using owner-reported measures, and the total number of dogs involved in the study was equal to 19. Additionally, although we agree with you that the accuracy of evaluation can vary a lot from person to person and a larger sample size is always preferable, the palatability in this study was based on voluntary acceptance intake and assessed according to objective rules that were independent from the owners’ personal judgment (e.g. the dog eats/refuses the spirulina tablets offered alone) and the same could be said for the main aspect of the health assessment (e.g. the dog vomits/does not vomit; the dog experiences diarrhea/does not).

2-The daily amounts of Spirulina provided to dogs and cats was fixed per category: the starting amounts were 0.4 g/d (= 1 tablet) for cats and small-size dogs, 0.8 g/d (= 2 tablets) for medium-size dogs, 1.2 g/d (=3 tablets) for large-size dogs. We then proceed to double them for week 3 and 4 and to triple them for week 5 and 6. Since the animals in each category had different body weights (for example, in the medium-size category the dogs weight varied between 11 and 25 kg), naturally the amount of Spirulina expressed as g per kg BW varied and it was a range. But this is true for every nutraceutical supplement on the market, and even for many drugs (and Spirulina is not a drug), both in veterinary and in human medicine. For example, in humans it is often suggested to use Spirulina at 1 g per day. That is equal to 16 mg x kg BW for a person that weighs 60 kg and 11 mg x kg BW for a person that weighs 90 kg. It’s inevitable that there’s going to be a range. Many veterinary supplementations on the market do not even consider the differences in body weight of dogs, and just recommend to use the same daily amount for any dogs. We believe our division in the categories based on body weight is the best way to proceed.

3-Our aim was not to determine the daily amounts appropriateness, but to verify if there was a correlation between increasing amounts and palatability and if the owners would notice side effects. We choose the initial daily amounts after reviewing the literature in humans and then proceed to double and triple those initial amounts to further test our hypothesis. We fully agree with you that to evaluate the objective effects of the supplementation blood samples must be taken and a placebo group must be used, but this was beyond the aim of the present study. The data from this study were very helpful to us as a foundation to then prepare a randomized controlled trial concerning the use of Spirulina in overweight dogs, where we did took blood samples at different time points. We are currently working on the manuscript concerning such trial and we are going to present the results at the next ACVIM forum in June.

4-We added this information in the intervention protocol in material and methods: The initial daily amount of Spirulina was selected after reviewing the literature concerning the use of Spirulina in humans, which is usually in the range of 1-4 g/day, equal to 0.01-0.06 g/kg/day [16-18].

5-We reported the p-value of each parameters assessed in the results (e.g. in lines 369, 384, 390, etc).

Thank you for the additional comments. We have made revisions accordingly, following your suggestions. A few clarifications:

3-we added that info in line 177 at the beginning of the chemical analysis chapter

4-the breed of the dogs and cats are mentioned in paragraph 3.2 and 3.3

9-the spectrophotometer reading wavelengths were from 350 to 750 nm; then, in that range, the absorbances were read at 664, 461, 615, 652 nm as it is described in the text.

Reviewer 2 Report

This article is a case study on the owners’ perception of the effect of increasing amounts of Spirulina. It is a very simple research with limited relevance due to the small sample size and its heterogeneity. The data have no statistical significance and can only be food for thought.

Nonetheless, the work is well-written and clearly presented. Therefore it can be published for those who want to continue research in this area. 

Author Response

Thank you for your review. We agree with your feedback; the purpose of this work was indeed to gather some initial information about the nutraceutical use of microalgae in pet nutrition, so more in-depth studies with stronger protocols could be then developed.

Reviewer 3 Report

All names of microorganisms must be write in cursive 

Author Response

Thank you for your review. We checked the manuscript in order to make sure all names of microorganism were written in cursive.

Reviewer 4 Report

  • Dear authors,

    Thank you for a well written manuscript of a novel subject. Pet nutrition and various feed additives and complementary feed materials have become very popular as the role of pets as family members has increased. This manuscript is thus very welcome in the scientific community despite its limitations (e.g. lack of control group, subjective parameters).

    Palatability is very important when it comes to animals as the product needs to be ingested voluntarily by the animal. In this study you have described as the best alternative that the owners offered the tablet as a whole directly to the mouth. In the palatability Guideline on the demonstration of palatability of veterinary medicinal products (EMA/CVMP/EWP/206024/2011), for assessing the acceptance of the test product, it could be offered in the following pre-determined order: first, it may be offered in an empty bowl or trough, or on the ground (depending upon species behaviour) to assess voluntary acceptance during one minute. Please describe in more detail how the tablets were offered in your study to avoid misunderstanding that the tablets were put directly to the back of the tongue of the animals by the owners. Voluntary uptake should be emphasized and how tablets were offered to the animals. 

    In the materials and methods section you have described in very detailed manner how each component of Spirulina was analysed (section 2.5 Chemical analyses) but I could not find a description of the composition of the study product ie did it contain anything else than dried algae and how was it manufactured, what was the origin of the algae.

    In the results section you have very detailed information about each parameter assessed - please consider some other form to present at least some of these results (e.g. tables of graphs) as currently it is quite exhaustive to read. Scratching and lacrimation was measured but I could not find a reason for these measurements. Does Spirulina increase lacrimation?

    You are using any different terms to describe that animal owners decided to withdraw their consent and discontinue the study - please consider only one term (discontinue) to avoid confusion about the reasons for discontinuation (unless this is the intention).

    Specific comments:

    row 15: please change the word medicine to feed as complementary feed is the term used, medicines are drugs

    row 22: in the simple summary you say that Spirulina in companion animals is still largely unexplored but here you state that it has not been investigated at all – which one is correct as in the introduction you refer to some dog and cat publications

    row 59: please consider rewording “among which” or add some verbs to the end of this sentence. Currently the meaning of “among which” is unclear.

    row 88: please consider rewording or clarifying this sentence: “a preventive analysis of the Spirulina used for the experimental period was considered of utmost importance before starting the trial and therefore a further objective of the present study.” What is meant by preventive analysis? Was it done before starting the trial? What were the results?

    row 102: only dog owners signed the informed consent? What about cat owners?

    row 109: please consider changing the order of words: “owners were instructed to provide Spirulina daily for”

    row 144: information of scratching and lacrimation was collected but no justification why

    row 272: please consider combining breeds with the same number that you do not have to repeat the word “one” or “two” etc

    row 298: please consider changing the word “castrated” to “sterilized” or to “neutered” (male animals are normally castrated, not female animals)

    row 322: please consider deleting the word “however”

    rows 325 & 328: please consider changing the word “basal” to “initial”

    rows 323, 326 & 328: several different terms used for discontinuation – please consider using only one term if you mean that the animals discontinued the study

    row 336: please consider changing the term “modality of ingestion” to e.g. “method of offering”

    row 413: please consider changing the word “revealed” to e.g. reported

    row 414: please consider deleting the words “on the contrary”

    row 428: please consider changing the word “assuming”

    rows 456 & 457: where does the word equal refer to? Do not understand this text.

    row 464: spelling mistake “After” should not be written with capital A?

    row 496: please consider reformatting this table as it is difficult to read in the current format

    row 517: spelling error in the word “toxology”, should be “toxicological”

    row 545: please explain in more detail what is meant by “sensory attributes”

    row 573: please consider deleting the words “in particular”

    row 593: please correct the word “cosmeceuticals” to cosmetics

    row 630: in the case of animals the word “feed supplement” is used

    row 641: lacrimation mentioned here for cats, nothing in the dog chapter, nor explanation why it was measured

Author Response

Thank you for your review, your valuable comments helped us substantially improve this paper.

To address your major concerns:

1-the tablets were indeed offered alone in an empty bowl as first choice to assess the voluntary acceptance, and only later the other methods were used; thank you very much for your feedback on this point, the previous expression we used was definitely misleading and not clear enough, probably because of a bad English translation of the terms we used in the protocol that was initially written in Italian.

2-we specified that the spirulina used in this study was just dried algae derived from the raceway pond at the cultivation plant of the Italian company that provided us with the product.

3-we agree with you that it was an exhaustive reading; we changed the way we presented some data adding 4 more figures.

4-we asked the questions about scratching and lacrimation because these symptoms are very easily detected by owners and are among the most frequent ones related to food intolerances and allergies. So our aim was to verify if Spirulina could have an effect on these parameters, in the same way we wanted to verify if Spirulina could have an effect on vomiting, diarrhea, etc. Based on the feedback of the owners presented in the results, it seems that spirulina does not cause alterations in scratching nor lacrimation. We further specify this point in the discussion.

5-we changed the terms to describe that animal owners decided to withdraw from the study and used just the term “discontinue” as you suggested.

Thank you for the additional comments. We have made revisions accordingly, following your suggestions. A few clarifications:

On row 22: we changed “the nutraceutical use of Spirulina is still largely unexplored” in “the use of Spirulina is still largely unexplored” in the simple summary, while leaving “the nutraceutical use has not yet been investigated” in the abstract. This is because among the 3 studies concerning the use of spirulina in pet nutrition already published, 2 were in-vitro studies, and one was an in-vivo studies were dogs consumed kibbles that contained Spirulina. To our knowledge, there are no studies concerning the use of Spirulina as a nutraceutical supplement.

On row 88: we changed “preventive” in “chemical”. We are referring to the chemical analysis (proximate analysis, fatty acid profile, amino acid profile, mineral, vitamin and pigment quantification).

Round 2

Reviewer 1 Report

No further comments are to be addressed

Reviewer 4 Report

Dear Authors,

thank you for revising the manuscript according to comments. Now it is much more easier to read.

I detected only one spelling mistake on row 347: the word "a" should be deleted